Low effect of young afforestations on bird communities inhabiting heterogeneous Mediterranean cropland

Sánchez-Oliver Juan S. 1 2 juansanchezoliver@cibio.up.pt juanssoliver@gmail.com
Rey Benayas José M. 1
Carrascal Luis M. 3
1 Departamento de Ciencias de la Vida, Universidad de Alcalá , Alcalá de Henares , Spain
2 Applied Population and Community Ecology Laboratory, CIBIO/InBIO-Universidade do Porto , Lisboa , Portugal
3 Departamento de Biogeografía y Cambio Global, Museo Nacional de Ciencias Naturales-CSIC , Madrid , Spain
Gandini Patricia
Electronic publication date: 2015 Dec 7
Publication date: 2015
Volume: 3
Electronic Location ID: e1453
Received 2014 Jul 29; Accepted 2015 Nov 8
Copyright: © 2015 Sánchez-Oliver et al.
Copyright year: 2015
Copyright holder: Sánchez-Oliver et al.
License: This is an open access article distributed under the terms of the Creative Commons Attribution License, which permits unrestricted use, distribution, reproduction and adaptation in any medium and for any purpose provided that it is properly attributed. For attribution, the original author(s), title, publication source (PeerJ) and either DOI or URL of the article must be cited.
License URL: https://creativecommons.org/licenses/by/4.0/

Keywords: Conservation status, Distance effects, Land use types, Pine plantations, Species richness, CAP

Funding: Spanish Ministry of Science and Education CGL2010-18312 CGL2014-53308-P Government of Madrid S2009AMB-1783, REMEDINAL-2 S2013/MAE-2719, REMEDINAL-3 Fundación Internacional para la Restauración de Ecosistemas Projects from the Spanish Ministry of Science and Education (CGL2010-18312 and CGL2014-53308-P) and the Government of Madrid (S2009AMB-1783, REMEDINAL-2 and S2013/MAE-2719, REMEDINAL-3) have provided or are currently providing financial support for this body of research. JSSO was supported by a contract from Fundación Internacional para la Restauración de Ecosistemas (www.fundacionfire.org). The funders had no role in study design, data collection and analysis, decision to publish, or preparation of the manuscript.

==============================
Afforestation programs such as the one promoted by the EU Common Agricultural Policy have spread tree plantations on former cropland. These afforestations attract generalist forest and ubiquitous species but may cause severe damage to open habitat species, especially birds of high conservation value. We investigated the effects of young (<20 yr) tree plantations dominated by pine P. halepensis on bird communities inhabiting the adjacent open farmland habitat in central Spain. We hypothesize that pine plantations located at shorter distances from open fields and with larger surface would affect species richness and conservation value of bird communities. Regression models controlling for the influence of land use types around plantations revealed positive effects of higher distance to pine plantation edge on community species richness in winter, and negative effects on an index of conservation concern (SPEC) during the breeding season. However, plantation area did not have any effect on species richness or community conservation value. Our results indicate that the effects of pine afforestation on bird communities inhabiting Mediterranean cropland are diluted by heterogeneous agricultural landscapes.

Introduction

A significant amount of abandoned cropland, low productive cropland and pastureland has been converted into tree plantations in the last few decades, and ca. 7% of forest land in the world are tree plantations at present (FAO, 2011). Different afforestation programs have contributed to the spread of such tree plantations at the regional level. For instance, the Common Agricultural Policy (CAP) has favoured the conversion of farmland into tree plantations in the European Union since 1992 by means of a scheme of aid for forestry measures in agriculture (EEC Council Regulation No. 2080/92), which has resulted in the afforestation of ca. 8 million ha to date (European Commission, 2013a; European Commission, 2013b). Further, afforested cropland is expected to increase in the near future in countries such as Spain (with more than 500,000 ha at present) due to subsidies to afforestation of extirpated vineyards (Spanish Agrarian Guarantee Fund, 2012).

Tree plantations pursue a number of environmental and societal services such as soil retention and carbon sequestration (Rey Benayas et al., 2007). However, they may have noticeable effects on biological communities. Thus, Bremer & Farley (2010) found that tree plantations are most likely to contribute to biodiversity when established on degraded lands rather than replacing natural ecosystems, and when indigenous tree species rather than exotic species are used. Similarly, a meta-analysis of faunal and floral species richness and abundance in timber plantations and pasture lands on 36 sites across the world concluded that plantations support higher species richness or abundance than pasture land only for particular taxonomic groups (i.e., herpetofauna), or specific landscape features (i.e., absence of remnant vegetation within the pasture matrix) (Felton et al., 2010).

Agro-ecosystems are important for maintenance of bird diversity in Europe, especially for species of conservation concern (BirdLife International, 2004a). However, populations of farmland bird species in Europe showed a decline of ca. 20% between 1990 and 2008 (see also Donald et al., 2002; Gregory et al., 2005; Butler et al., 2010; Scholefield et al., 2011; Guerrero et al., 2012). Cropland afforestations in southern Europe, which are mostly based on coniferous species, may cause severe damage to open habitat species, especially ground-nesting birds, many of which are of conservation concern in Europe (European Bird Census Council, 2010). These negative effect are mostly due to the replacement of high quality habitat and increasing risk of predation (Santos et al., 2006; Caplat & Fonderflick, 2009; Reino et al., 2009; Reino et al., 2010; Voříšek et al., 2010; Butler et al., 2010). For instance, an assessment of nest predation rates on open farmland habitat adjacent to young tree plantations in central Spain resulted in 94.2% of artificial nests that were predated three weeks after the start of the experiment (Sánchez-Oliver, Rey Benayas & Carrascal, 2014a). Nonetheless, these tree plantations may increase local species richness close to them as they are suitable habitat for forest generalist and ubiquitous species, although they are not adequate habitat for forest specialists in their current state of maturity (Rey Benayas, Galván & Carrascal, 2010; Sánchez-Oliver, Rey Benayas & Carrascal, 2014b).

In a worryingly frame of open farmland species declines that, despite the efforts, is not being reverted, this study aims at investigating the effects of young (<20 yr) tree plantations on bird communities inhabiting the adjacent open farmland habitat in a Mediterranean landscape mosaic located in central Spain. Specifically, we hypothesized that distance to and area of tree plantations will affect species richness and conservation value of the bird communities because they will (1) attract only forest generalist and ubiquitous species of low conservation value and (2) have a detrimental effect on bird species that are characteristic of open farmland habitat, particularly for ground-nesting species that are of high conservation value. We predict that the effects of tree plantations may depend on land use type around them and that they will be more negative for open farmland birds in homogeneous herbaceous landscapes (Sánchez-Oliver, Rey Benayas & Carrascal, 2014b). We predict as well that these effects will be most noticeable in the breeding season than in winter due to territorial behaviour of birds and competition to avoid less favourable habitat (Morgado et al., 2010; Delgado et al., 2013).

Methods

Study area

Field work was carried out on open farmland adjacent to afforested cropland located in Campo de Montiel (La Mancha natural region, Ciudad Real province, southern Spanish plateau, UTM 30 S 469411 4289409; Fig. 1). The study area spreads on ca. 440 km2 with altitude ranging between 690 and 793 m a.s.l. The climate is continental Mediterranean with dry and hot summers and cold winters. Mean annual temperature and total annual precipitation in the area during the last 30 years were 13.7 °C and 390 mm, respectively (Agencia Española de Meteorología, 2012). These figures were 15.8 °C and 362.9 mm in 2012, when our bird surveys took place (Junta de Castilla-La Mancha, 2013).

Figure 1 Study area map.

Low effect of young afforestations on bird communities inhabiting heterogeneous Mediterranean cropland Location of the study area in central Spain within the Ciudad Real province and distribution of the young forest plantations (in black) and transects (grey) on adjacent cropland that were investigated in this study.

The area is a representative mosaic of different crops, pastures and natural or introduced woody vegetation that are characteristic of large areas in Mediterranean landscapes. Cropland was mostly herbaceous crops (24%, principally wheat and barley) and permanent woody crops (22% olive groves and 18% vineyards). Natural vegetation consisted of Holm oak (Quercus rotundifolia L.)  woodland and riparian forests that have been mostly extirpated from this region. Until 1992, woodland cover was restricted to open Holm oak patches, usually grazed by sheep and goats. Major land use changes in the last years in Ciudad Real province were vineyard extirpation (17,620 out of 24,347 ha in total between 2008 and 2011; Government of Castilla-La Mancha, 2013, unpublished data) and abandonment of herbaceous cropland and subsequent afforestation (38,135 ha between 1993 and 2013, 1.9% of total area of Ciudad Real province; Government of Castilla-La Mancha, 2013, unpublished data). These afforestations are of small area (Table 1) due to property size and noticeably dominated by Aleppo pine (Pinus halepensis Mill.) alone or mixed with Holm oak.

Table 1 Charactersitics of tree plantations.

Mean, standard deviation (sd) and range (min/max) of area of tree plantations and land-use categories in 1-km × 200-m 80 transects on farmland habitat adjacent to the 40 tree plantations (two transects per plantation) that were surveyed in Central Spain.

	Mean	sd	min	max	
Area of tree plantation (ha)	5.8	6.6	1.3	36.5	
Cover of the tree layer (%)	38.8	25.7	1.7	100.0	
Average pine height (m)	3.6	1.5	1.0	7.2	
Average trunk diameter of pines (dbh cm)	13.3	6.5	4.0	33.2	
Streams and rivers (% cover)	0.3	0.9	0.0	6.4	
Roads and rural tracks (% cover)	1.7	1.6	0.0	7.7	
Olive groves (% cover)	14.2	21.5	0.0	94.5	
Scattered buildings (% cover)	0.3	1.0	0.0	7.3	
Tree plantations (% cover)	0.8	2.5	0.0	22.7	
Natural woodland (% cover)	0.1	0.8	0.0	9.5	
Fruit and dried fruit groves (% cover)	0.2	2.1	0.0	26.6	
Waste lands (% cover)	1.7	4.6	0.0	31.8	
Pastures (% cover)	6.5	15.8	0.0	99.2	
Dry herbaceous cropland (% cover)	40.4	32.8	0.0	100.0	
Vineyards (% cover)	33.9	32.2	0.0	100.0	

Selection of tree plantations for bird survey at adjacent farmland habitat

First, all tree plantations in the study area were located using both orto-photos (Geographic Information System of Farming Land, 2010; hereafter SigPac) and Google Earth®, and were later verified in the field. We found 99 tree plantations on former cropland that took place in 1992 or later. Tree plantations <1 ha were directly discarded. In addition, a target tree plantation had to be placed at least 2-km away from another plantation in the transect direction to avoid that surveyed birds associated to open farmland adjacent to a given tree plantation were affected by another tree plantation. Following these criteria, we finally selected 40 tree plantations to assess bird community on farmland adjacent to tree plantations. We measured the area of every tree plantation (Table 1) using ArcGIS 10.0 (ESRI Inc., Redlands, CA, USA). As they were young, the tree canopy was little developed (mean tree cover = 38.8% ± 25.7%, mean tree height = 3.6 m ± 1.5 m, and mean dbh = 13.3 cm ± 6.5 cm).

Bird survey

Bird census were carried out in winter (January and February) and breeding season (April and May) of 2012 to assess the wintering and breeding bird communities, respectively. Census method consisted of outward line transects of 1,000-m length with belts of 100-m at each side of the observer and initiated at the tree plantation edge (Bibby et al., 2000; Gregory, Gibbons & Donald, 2004). Two census-transects for each plantation and season were carried out in different days, one in the morning between sunrise and three hours later and one in the evening two hours before sunset (80 transects in total). The two transects from each tree plantation spanned on different directions that were established a priori to meet the criterion used for selection of tree plantations (see above). They were walked at an average speed of 2.5 km h−1. We noted and geo-localized the presence of every bird except those that were over-flying the census area (i.e., distance to the tree plantation edge and situation with respect to the transect progression line; Table S1). All censuses were conducted by the same well trained field ornithologist (JS S-O) on windless and rainless days.

The European endangered status of each species was obtained from BirdLife International (2004b) using the Species of European Conservation Concern (SPEC index) scores. This index uses four categories: SPEC 1 (global conservation concern), SPEC 2 (concentrated in Europe and with an unfavourable globally threatened or near threatened conservation status or data deficient), SPEC 3 (not concentrated in Europe but with an unfavourable conservation status), and Non-SPEC (favourable conservation status). We assigned a value of 4 to species that were included in the Non-SPEC category. Finally, we used a transformed SPEC index by subtracting the SPEC value of each surveyed species to 5 in order that species of highest conservation concern attained the greatest value (4), whereas the species of lowest conservation concern attained a value of 1 (De la Montaña, Rey Benayas & Carrascal, 2006). The average values of the transformed SPEC index were calculated considering the recorded species occurrence in each transect (Table S1).

To have a reference of the avifauna that colonizes farmland habitat in the studied region, for comparison with our bird survey, we used (1) the species list (47 species) of the common farmland bird indicator for Southern Europe (European Bird Census Council, 2010) and the list (36 species) of common farmland bird index (Directorate-General for Agriculture and Rural Development, 2012), and (2) the mean density of breeding species found at the habitat categories labelled as (a) dry arable lands, (b) vineyards, (c) olive groves, (d) agricultural mosaics with woody cultivations, and (e) pastures within the Mesomediterranean region of Central Spain obtained from Carrascal & Palomino (2008) (Table S1).

Land use types

We measured percentage of land use types in all 80 transects on farmland habitat where bird survey took place using ArcGIS 10.0 (ESRI Inc., Redlands, CA, USA). The length and width of transects make them representative samples of land use types in the studied landscape. Land use types were identified by means of land use layers taken from SigPac (Geographic Information System of Farming Land, 2010) and verified in the field. We initially distinguished 21 land use types that were aggregated into the following eleven categories for statistical analyses according to their larger covers in the study region (i.e., avoiding those habitat categories of very low representativeness): streams and rivers, roads and rural tracks, olive groves, scattered buildings, tree plantations other than young cropland afforestations (i.e., on non-arable hills), natural woodland, fruit and dried fruit groves, waste lands (unproductive, dump areas), pastures (semi-natural grasslands), dry herbaceous cropland (mostly cereals), and vineyard. The percentage of area occupied by each land use type across transects is shown in Table 1.

Statistical analyses

Water birds (e.g., Common Sandpiper Actitis hypolecus, Mallard Anas plathyrhinchos, and Grey Heron Ardea cinerea), aerial feeders (European Bee-eater Merops apiaster and the Hirundinidae family species), and raptors were not considered in data analyses as the census method we used does not accurately estimate the occurrence of these species. Species richness and SPEC score of the whole bird community, including forest species, were analysed by means of Generalized Linear Models based on a Poisson distribution (with the log-link function) with distance to tree plantation and plantation area as target predictors. Distance to tree plantation edge was treated as a dummy variable, i.e., 0 for close or <400 m vs. 1 for away or 600–1,000 m. We used the 400 m and 600 m thresholds for distances “close to” and “away from” plantations, respectively, as representative of the landscape scale addressed in this study which allow for analysing differences in bird responses between specialist and generalist farmland species (the farther points from forest edge were situated at 1,531 ± 1,201 m distance in Reino et al., 2009). The 400–600 m distance was treated as a buffer zone and not considered in the analysis. Area of tree plantation was included as a continuous covariate (in logarithm). The height of trees was included in previous analyses but it did not have any significant effects and thus it was eliminated in final analysis. As land use type may affect abundance of bird species around plantations, we included in the models the cover of six land use categories with a percentage higher than 1% as control covariates (namely, roads and rural tracks, olive groves, waste lands, pastures, dry herbaceous croplands and vineyards). GLMs were carried out with Gretl (release 1.9.5, http://gretl.sourceforge.net/). Statistical significance of the predictor variables was calculated using quasi-ML standard errors. We also tested for homogeneity of slopes of plantation area in the close and away transect sectors in a posteriori regression analyses.

Results

We detected a total of 3,643 individuals belonging to 47 species in winter and 1,149 individuals belonging to 37 species in the breeding season at our 80 1-km transects (Table S1). Thirty two species were included in the Non-SPEC category—least conservation concern—, 12 in the SPEC 3, nine in the SPEC 2, and two in the SPEC 1—highest conservation concern (Table S1).

Models revealed significant effects of distance to tree plantation edge on community species richness in winter (i.e., communities away from the plantations were ca. 30% richer in species than those close to plantations) but not in the breeding season (Table 2). The plantation area term did not have any effect on species richness in winter or breeding season. The effect of land use types was not significant.

Table 2 Results of Species Richness and SPEC.

Species richness and the average transformed SPEC index related to conservation concern (in an inverse scale from 1-safe to 4-highly threatened) of the bird fauna inhabiting areas close to (0–400 m) and away (600–1,000 m) from forest plantation edges, in winter (A) and in the breeding season (B). Figures are mean ± sd. The regression coefficient and p-value of the effects of distance to plantation edge and plantation area (log-transformed) were obtained using generalized regression models that compare close vs. away (as a dummy variable: 0-close, 1-away) controlling for the effects of land use type (see ‘Methods’ for more details).

	Close	Away	Distance	Area of plantation	
	Mean ± sd	Mean ± sd	Coeffic.	p-value	Coeffic.	p-value	
A. Winter							
Species richness	2.76 ± 2.06	3.56 ± 2.25	0.127	0.015	−0.045	0.570	
Transformed SPEC index	1.39 ± 0.68	1.48 ± 0.64	0.054	0.337	−0.028	0.694	
B. Breeding season							
Species richness	3.20 ± 1.88	3.01 ± 1.87	0.000	0.996	−0.041	0.490	
Transformed SPEC index	1.86 ± 0.67	1.60 ± 0.72	−0.116	0.037	0.011	0.879	

Distance to tree plantation edge showed a significant effect on the SPEC index during the breeding season (i.e., 19% higher index close to tree plantations), but not in winter. Plantation area did not have any effect on the SPEC index in either season (Table 2).

Discussion

Overall, we found that young tree plantations established on former cropland in a Mediterranean mosaic located in central Spain had (1) a low detrimental effect on bird species richness in winter and (2) a marginal positive effect on conservation value of bird communities at adjacent open farmland habitat in the breeding season. Our results and their conclusions have all limitations of a single year study.

Previous studies on the effects of tree plantations in open habitat bird species have mostly found negative effects, particularly for the most specialized and of more conservation concern species (Shochat, Abramsky & Pinshow, 2001; Santos et al., 2006; Reino et al., 2009; Reino et al., 2010; Reino et al., 2013; Morgado et al., 2010). For instance, Fonderflick, Besnard & Martin (2013) found that the abundance of open-habitat birds decreased significantly in the vicinity of edges, this negative response extended within 150 m from the edge, and the effect was disproportionately higher in open-habitat species with high conservation concern. We found this detrimental effect for species richness in winter, but this result refers to all species of the community and not only to open habitat species (Table 2A). These simple and monospecific tree plantations would provide scarce food resources in winter, causing bird communities avoid them for foraging in more productive crops like olive groves or vineyard (Ries et al., 2004; Myczko et al., 2013). In turn, the negative response of steppe birds but not of generalist farmland species to edge contrast could reduce the number of species close to younger tree plantations, since gregarious behaviour of steppe birds in winter would enable these species to move away from them (Reino et al., 2009; Delgado et al., 2013).

Conservation status concern of the bird assemblage in the breeding season was higher at close distance to the tree plantation edge and was not affected by the area of tree plantations (Table 2). The small size of the plantations (5.8 ha in average) together with the little development of some of them (e.g., tree cover of 1.7%, Table 1) may produce detrimental effects only at very short distances from them (e.g., <150 m, Fonderflick, Besnard & Martin, 2013; Sánchez-Oliver, Rey Benayas & Carrascal, 2014a). The plantation area may modify the magnitude of the distance effect, which should be higher in larger patches than in small ones. As we noted above, scant development of these plantations and low edge contrast may explain their little effect on bird communities, but tree growth may increase the negative effect in open farmland birds in the future, especially for conservation concern species (Reino et al., 2009).

Further, these plantations may mirror remnants of natural or semi-natural woody vegetation such as woodland patches and hedgerows that may be even beneficial for some farmland bird species (e.g., buntings), as they offer opportunities for forage, refuge and breeding (Concepción & Díaz, 2010; Concepción & Díaz, 2011; Morgado et al., 2010; Batáry et al., 2012). Importantly, the hypothesized detrimental effect of the tree plantations seems to be diluted by the high heterogeneity of the landscape mosaic and the important proportion of woody crops such as olive groves (Table 1) (Tryjanowski et al., 2011; Myczko et al., 2013). In agreement, other studies have shown that landscape heterogeneity is a relevant factor affecting the occurrence and abundance of farmland birds (Morales, García & Arroyo, 2005; Batáry, Matthiesen & Tscharntke, 2010; Batáry et al., 2011; Flohre et al., 2011; Concepción & Díaz, 2011; Sánchez-Oliver, Rey Benayas & Carrascal, 2014a).

We conclude that distance to, but not area of, young pine plantations established on former Mediterranean cropland exert a low detrimental effect on bird species richness but not in the overall community conservation status at open farmland, which seems to be diluted by the high heterogeneity of the landscape. We recommend long-term assessments of afforestations in agricultural landscapes to fully understand their effects on farmland bird community and, consequently, propose management measures to reduce or avoid their possible negative impacts on biodiversity, particularly on ground-nesting birds.

Supplemental Information

Supplemental Information 1 Supplementary Material

Click here for additional data file.

Additional Information and Declarations

Competing Interests

Author Contributions

Data Availability

The authors declare there are no competing interests.

Juan S. Sánchez-Oliver conceived and designed the experiments, performed the experiments, analyzed the data, contributed reagents/materials/analysis tools, wrote the paper, prepared figures and/or tables, reviewed drafts of the paper.

José M. Rey Benayas conceived and designed the experiments, contributed reagents/materials/analysis tools, wrote the paper, prepared figures and/or tables, reviewed drafts of the paper.

Luis M. Carrascal conceived and designed the experiments, analyzed the data, wrote the paper, reviewed drafts of the paper.

The following information was supplied regarding data availability:

The research in this article did not generate any raw data.

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
