# Peer review of "Low effect of young afforestations on bird communities inhabiting heterogeneous Mediterranean cropland"

_PeerJ, doi:10.7717/peerj.1453_

## Round 0.1 · original submission · Major Revisions

· Academic Editor

Major Revisions

The article investigates the effect of surface and distance of young pine plantations on bird communities. The paper is well written but including some reviewer comments should be done before its final acceptation.
1. You need to describe afforestation
2. Hyphothesizes and methods should be clarified (follow Rev.2).
3. Authors should include plantation sizes and discuss if it affects or not your results
4. Explain which criteria did you use to chose the distances in the paper.. If authors were chosen others the results will be other? You should discuss it
5. Follow carefully all comments made by Rev. 4
6. Explain why some species will be seriously impacted or less impacted
7. Why do you think distance have possitive effect during winter? Is there a Threshold level of afforestation for it? Follow Rev. 5.
Also some other specific comments are included in reviewer reportings. Please considere all of them, specially Rev 1 and 2.

Reviewer 1 ·

Basic reporting

No comments about the article writing and structure.
The subject is properly introduced but some figures about arable land afforestation in Spain would be welcome, to understand the magnitude of the process in the study region. In this same direction, some more data about the landscape structure at the study area (i.e. total cover percentages of plantations and main land uses, not just those around transects) would help readers not familiar with Mediterranean landscapes.
I have found an issue with Table S1: line 228-230, “Little Bustard [...] associated to larger plantations “, there is no information about plantation size, so one cannot see how the species “associates” to larger plantations.

Experimental design

Research question: although, as stated in the introduction, the main negative effect of cropland afforestation on farmland birds is expected to be habitat loss, investigating how these communities interact with plantations it is of interest. Nevertheless, as it is suggested in the discussion, 400 m. may be too far to be close.

Methodology: still, methodology is quite solid and well described. Target species are carefully selected and sampling effort is remarkable. I have just one minor concern: among land use classes, I can see one “afforestation” category which I would like to know how does it differ from studied plantations.

Validity of the findings

As said before, maybe those two distance categories may not be adequate as one can infer from previous literature cited in the discussion section. That could be the reason behind the surprising results (very low effect of tree plantations on open land bird communities). I wonder if authors tested different distance categories.
All the same, in the discussion, authors refer to plantation structure as one possible explanation of the obtained results. Well, having those data, maybe it would be interesting to test whether plantation structure has some effect on farmland bird communities.

Additional comments

In general, this study is a valuable contribution to the study of the effect of land use change on farmland bird communities, in a worryingly frame of populations declines that, despite the efforts, are not being reverted.

Reviewer 2 ·

Basic reporting

This paper about the impact of afforestation on bird communities is interesting. However, several questions and comments need to be solve to improve the quality of the mansucript. In priority I think that hypothesizes and methods should be clarified.

>>Abstract
L20 This is not clear. You wrote "We hypothesize that pine plantations with larger surface, and areas at shorter distances from plantations, would result in lower bird species richness"
but you contradict yourself L65 : "tree plantations are a suitable habitat for forest generalist and ubiquitous species, which may lead to an increase in local species richness"

L24 (and in the title) you wrote "has an overall low detrimental effect on bird species that are characteristic of open farmland habitat" but I am not sure that you exclusively work on open land species (in the two metrics you used, species richness and SPEC index, forest species are included). See my comments below

>> Introduction
L 52 It is strange to cite the "Directorate-General for Agriculture", please just use the references L 55

L59 "This negative effect..."

L65 This type of habitat cannot also harbour forest specialists ? In Sanchez-Oliver et al. 2014, you mentioned in the discussion that these plantations offer habitats for generalists but not for forest specialist probably due to the biogeographical basis of you Mediterranean region. I think you shouldn't easily generalize this result.

L72 Do you mean species richness of all birds ?

L73 Please modify and adjust your hypotheses because you don't work specifically on open farmland species and ground nesting birds

L76 detail a bit these two hypotheses :
-In which way surrounding land uses can influence the afforestation effect ? Are there specific processes ? Complementation ?
-Why will the tree effects be more noticeable during territorial behaviour of birds ? You should develop and explain why you expect different results between seasons. Give a few references.

>>Methods
L90 please replace "herbaceous crops" by "cereals"

L94 Please quantify a bit the abandoned land to give a precise idea of agricultural evolution in this region. Quantify also the subsquent afforestation (total area, patch number, ...)

L104 Give a reference

L 133 Why do you consider the Non SPEC species (with a value of 1) in the calculation of the SPEC index ? Why not to remove these species in this index ?

L136 How did you calculate the SPEC index at the transect scale ? Did you consider the frequence or abundance of each species (Table S1) ? Please give the formula to be clearer.

L153 Is there a difference of between the land use "afforestation" and the 40 tree plantations? What is the plant composition ? Please detail a bit the following categories: semi natural
woodland, waste lands, pastures (a synonymous of natural grassland ?), dry herbaceous cropland. This is not very clear if we don't know very well your study system.

L160 Add a sentence to explain why you didn't consider these species.

L160 Please explicit that you work on the total species richness (if I understand) including also forest species (e.g. Phylloscopus collybita, Sylvia atricapilla). If I understand, your SPEC index include open land species but also species associated to forest habitat (e.g. Lophophanus cristatus, Streptopelia turtur)
You announce in the title that you work on open land species, I am surprise you did not consider specifically metrics about specialists of open habitats, for exemple species richness of ground nesting birds.

L161 It is necessary to use a Poisson distribution to model species richness which is a discrete variable.
Did you examinate spatial autocorrelation in residuals of GLM ? This is important because it may bias parameters and can increase type I error (Dormann et al. 2007 Ecography).

L162 Explain why you didn't treat the distance to tree plantation edges as a continuous variable but as a dummy variable. Because it was difficult to estimate exactly bird positions during sampling ? Why didn't you consider a 400m-600m category ? How did you consider birds that were located in this area ?

L172 Delete this sentence, you show that plantation area has no effect.

>> Discussion

L191 All cited studies are not about tree plantations but also on land abandonment and spontaneous afforestation?

L198 Your result is about the total richness, including forest species (e.g. Phylloscopus collybita, Turdus viscivorus, ...Table S1) or just on open land species. Fonderflick's conclusion is specifically on open land birds, results are not really comparable.

L220 This conclusion is not supported by the results. Modify the sentence.


Figure 1 Can you show all tree plantations (even non-selected plantations) to give an idea of the distribution of these plantations in landscapes

Table 2
-Please harmonize "Transformed SPEC index" and "SPEC index (inverse of)"
-Move down "Winter" in the same level that "Breeding season"

Table S1
-modify "Entire"
-I don't understand why for certain species, when summing frequencies of close and away segments the result is higher than in the entire transect (e.g. Pica pica, Galerida cristata)

I hope these comments will help you to improve your manuscript.
Good luck

Experimental design

See the Basic reporting section

Validity of the findings

See the Basic reporting section

·

Basic reporting

This manuscript investigates the effects of surface and distance from young pine plantations on bird communities breeding and wintering at the adjacent open farmland habitat in central Spain. The authors conclude to detrimental effect on species richness in winter and positive effect on conservation value of bird assemblage during breeding season. The study is interesting for several reasons. First, because the study of potential impacts of afforestation on farmland birds is an interesting topic in ecology and conservation. Second, because the manuscript can help in the planning of future (and predictable) afforestation programs in high-value areas from the point of view of biodiversity. Technically, the main frame of the analysis is well conducted, although I identify some points that could be clarified to improved manuscript. I only have a few rather comments considering minor changes to the manuscript, hoping that they will be constructive enough.

Methods

L162-164: Why you don't use distance to plantation edge as continuous predictor? Please, state why you use the cut-off point of 400 m in this approach (with the consequent loss of information).


Results

It is possible that distance to tree plantation affect species richness or the SPEC index in some habitat types but not in others? Maybe you can easily explore this possibility with your analysis and give some more detailed information on it (perhaps important from the point of view of conservation strategies addressing major threats for each habitat type).

Discussion

Please, state briefly the potential limitation of main conclusions due to a 1-year study

L203: There should be something wrong in table S1, at least in my word document. Data are difficult to read (to assign some data to the correct row) for some species (Passer domestics and Tetrax tetrax, for example).

L217: While your data show a lack of significant relationship between plantation area and species richness or SPEC index, I would be much more cautious on everything that relates with this variable. If you detect a detrimental effect of distance to pine plantation, then the surface of plantation necessarily affect too because the larger the patch the greater the perimeter and, consequently, the surface of the buffer (e.g. >400m) around each patch that was negatively influenced. Therefore, the “magnitude of the distance effect” (how many birds are finally affected, in absolute numbers) should be larger in larger patches than in small ones.


Finally, the results let me wondering about the bird richness and the SPEC index that was in the area before the afforestation, so the –overall- impact of plantations in agrosystems may be even worse than can be detected at a given time later after the management.

Experimental design

No comments (see above)

Validity of the findings

No comments (see above)

Additional comments

No comments (see above)

Reviewer 4 ·

Basic reporting

The manuscript gave detail descriptions including selection of sampling sites, land using type, and method of surveying birds. The questions giving in introduction were also interesting. But, most data were species richness, and it seems the result could not answer the questions. So, it is very necessary to greatly improve in the following points.
1) What kinds of species will be seriously impacted, or less impacted? So, you need to category them, such as ground-nesting species etc. Which are generalist/ubiquitous species? It is better that you will give a table with these detail informations.
2) It is necessary to give detail discussion why did distance have positive effect in winter? It is very weak in discussion in present version.
3) You have not used the data in Table 1. So, you would not explain what reasons reduced in your results?

Experimental design

No comments

Validity of the findings

No comments

Additional comments

In general, the manuscript need to greatly improve.

Reviewer 5 ·

Basic reporting

No Comments

Experimental design

No Comments

Validity of the findings

The authors' overall conclusion is that cropland afforestations are detrimental for birds and should not be considered in habitat restauration policies. However, they also found some positive effects on a species of high conservation value, i.e. the Little Bustard, during the breeding season. Is there any way of executing pine plantations at a minimum while keeping the positive effects on Little Bustard populations? The authors should suggest ways to get a threshold level of afforestation that allows obtaining those positive effects on breeding populations of birds.

Additional comments

In addition to my suggestion for discussing a strategy that minimizes the level of afforestation while keeping positive effects on some species, the authors should explain in the Introduction what is the novelty in the aims of this study, as other studies have previously shown detrimental effects of pine plantations on the populations of birds of high conservation value in the same study area.

---

## Round 0.2 · accepted · Accept

· Academic Editor

Accept

Hypotheses and methods have been clarified. Authors also explain which criteria did they use to chose the distances in the paper. The suggested figures have been provided Afforestation is described and plantation sizes included. Also they discuss if plantation sizes affects or not the results. The differences of impact among species is explained. Minor changes were also included.